# Emission of Particulate Inorganic Substances from Prescribed Open Grassland Burning in Hirado, Akiyoshidai, and Aso, Japan

Satoshi Irei [1,*], Seiichiro Yonemura [2], Satoshi Kameyama [3], Asahi Sakuma [4] and Hiroto Shimazaki [4]

[1] Department of Environment and Public Health, National Institute for Minamata Disease, 4058-18 Hama, Minamata 867-0008, Japan
[2] Faculty of Bioresource Sciences, Prefectural University of Hiroshima, 5562 Nanatsuka, Shobara 723-0023, Japan
[3] Biodiversity Division, National Institute for Environmental Studies, 16-2 Onogawa, Tsukuba 305-8506, Japan; kame@nies.go.jp
[4] Department of Civil Engineering, National Institute of Technology, Kisarazu College, 2-11-1 Kiyomidaihigashi, Kisarazu 292-0041, Japan
* Correspondence: satoshi_irei@env.go.jp

**Abstract:** Biomass burning is one of the largest sources of particulate matter emissions globally. However, the emission of particulate inorganic species from prescribed grassland burning in Japan has not yet been characterized. In this study, we collected total suspended particulate matter from prescribed grassland burning in Hirado, Akiyoshidai, and Aso, Japan. The collected filter samples were brought to the laboratory, and water-soluble inorganic components were analyzed via ion chromatography. The measurement results showed high excess concentrations of potassium, calcium, and magnesium, and these substances were highly correlated, which agreed with previously reported findings. In contrast, the concentrations of sodium, chloride, nitrate, and sulfate were insignificant, even though their high concentrations were reported in other biomass burning studies. Among these low concentration substances, a high correlation was still observed between sulfate and nitrate. It is possible that the low concentrations of those species could have been biased in the measurements, particularly as a result of subtracting blank and background values from the observed concentrations. Building up more data in this area may allow us to characterize the significance of domestic biomass burning's contribution to inorganic particulate components in Japanese air, which may consequently contributes to better understanding of adverse health effect of airborne particulate matter.

**Keywords:** aerosol; PM; air pollution; biomass burning; inorganic composition; grassland

## 1. Introduction

Airborne particulate matter, or PM, is an important and key component of the atmosphere in terms of the radiative forcing of the Earth [1]. In the last decade, the adverse health effects of PM have attracted great attention [2]. PM also has a role in the material cycle in the biosphere; it delivers inorganic PM components to locations distant from their origins. The main PM sources are ocean spray, sand and soil dust, and anthropogenic emissions. But in some local and regional cases, other major sources might exist, such as biomass burning. To date, inorganic PM components from local sources in Japan have not been fully understood. From the point of view of air quality policy, chemical characterizations of locally emitted PM are important not only to compile basic knowledge on the emission inventory of PM components, but also to better chemically characterize long-range pollutants, which are often mixed in the air of western Japan. By characterizing domestic sources of biomass burning, we will have a clearer goal that governmental and, in turn, international policy on air quality can achieve to control PM pollution.

Crutzen et al. [3] first reported the significance of biomass burning as a source of airborne chemical components in the atmosphere. Intensive field studies on wildfires,

biofuel burning, prescribed fires, etc., have been conducted, and it is known that approximately 38.3 Tg of PM [4] and 8% of total gaseous mercury [5] emitted to the atmosphere are from biomass burning origins. What about inorganics? Potassium is known to be an inorganic marker substance for biomass burning [6], but it is also derived from the ocean. Typically, potassium originating from biomass burning is estimated by subtracting the amount of potassium of ocean origin from the total potassium concentration using sodium concentration. In this estimation, we assume the origin of sodium is 100% from the ocean. However, a report conducting inorganic ion analysis implied that this is not likely true in some cases [7], and biomass burning can be a source of ions that have previously been believed to be of marine origin.

Prescribed open grassland burnings, called Noyaki or Yamayaki in Japanese, are traditional events in some locations in Japan, such as Hirado, Akiyoshidai, and the Aso region. It is said that Noyaki in the Aso region has been practiced since ten thousand years ago for the conservation of grasslands [8]. Contradictory to the biomass burning studies performed overseas, domestic biomass burning has not gained attention because large biomass burning events are rare in Japan. However, the traditional prescribed fires in these regions in February and March are the largest domestic open biomass burning events, and their emissions and significance in the Far East Asian region have not been evaluated to date.

The objective of this study was to better understand the association of ubiquitous inorganic substance emissions with domestic biomass burning. Taking advantage of such traditional events in our region, we conducted field studies on prescribed open grassland burning. In total, 15 suspended particulate matter (TSP) filter samples were collected during Noyaki and 10 water-soluble ionic species in those samples were analyzed and evaluated if those ionic species were emitted from the prescribed grassland burning.

## 2. Materials and Method

*Sample Collection*

Field studies were conducted during Noyaki in Akiyoshidai, Hirado, and 5 municipalities in the Aso region (Aso city, Minamioguni town, Oguni town, Nishihara village, and Ubuyama village), Japan (Figure 1). During these events, sampling devices, pumps, and batteries were installed on cars, and the plume was drawn through sampling filters positioned in the windows. The success of ground sampling strongly depends on wind directions and sampling location. Therefore, the cars were re-located frequently so that we could sample the gases in the plumes of Noyaki as directly as possible.

PM sampling during Noyaki was conducted together with total gaseous mercury sampling, which has been described elsewhere [9–11]. Briefly, 47 mm o.d. quartz fiber or PTFE-coated glass fiber filters (Tissuequartz or Emfab, Pall Corp., Part Washington, NY, USA) were used for PM collection. No pretreatment was performed for PTFE-coated glass fiber filters, but pretreatment of baking at 773 K for 24 h was performed for quartz fiber filters. Those filters were installed on open-face filter holders (Innovation Nilu AS, Kjeller, Norway), and the holders were attached to the head of large-volume fast gaseous mercury sampling tubes [9]. The sampling flow rate was mostly 80 L min$^{-1}$, but once in a while it fell to 60 L min$^{-1}$ due to pressure drops caused by filters being highly loaded with PM. Those highly loaded filters were replaced with new ones as soon as the pressure drops were noticed. Approximate flow rate correction was performed for those samples. Sampled air volumes ranged from 3.0 m$^3$ to 17.5 m$^3$. In total, 15 biomass burning samples were collected (Table 1). Beside Noyaki sampling, background air sampling (sampling during normal days; we hereafter denote those samples as BKG) was carried out 4 times, and 7 travel blank filters (the same sampling and analytical procedure with only a few seconds of air sampling at the sites; we hereafter denote those samples as BLK) were collected (Table 1). Filter samples were brought to the laboratory and stored in a freezer (263 K) until the samples were analyzed.

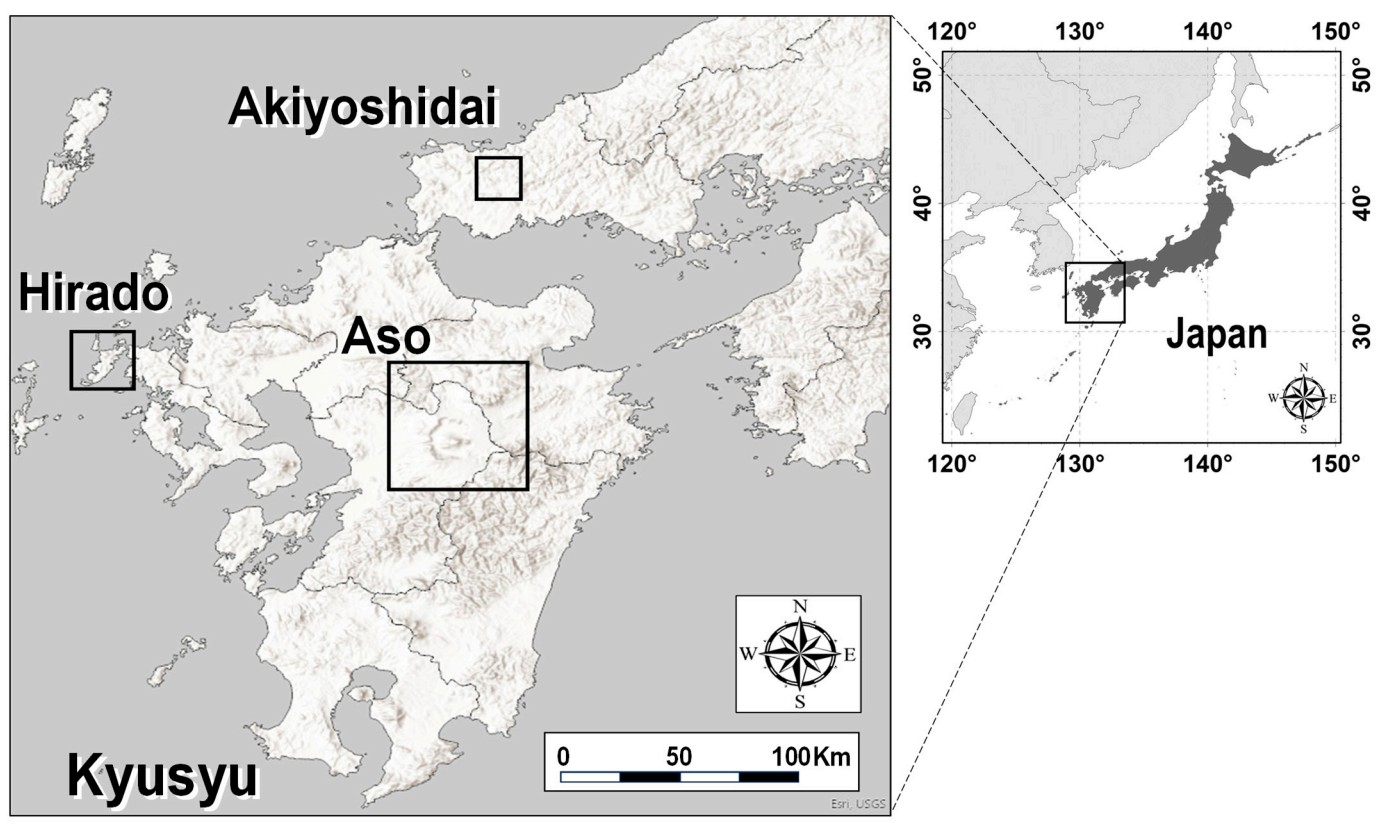

**Figure 1.** Map of Akiyoshidai, Hirado, and Aso.

**Table 1.** Summary of collected filter samples.

| | Year and Date | Sampling Time (min) | Flow Rate (L min$^{-1}$) | Type of Filter | Air Volume (m$^3$) |
|---|---|---|---|---|---|
| 2023 Hirado | 5 February 2023 | 93 | 80 | Quartz fiber | 7.4 |
| 2023 Akiyoshidai | 23 February 2023 | 122 | 80 | Quartz fiber | 9.8 |
| 2023 Aso_Nishihara | 26 February 2023 | 68 | 80 | Quartz fiber | 5.4 |
| 2023 Aso_Minamioguni 1 | 5 March 2023 | 92 | 80 | Quartz fiber | 7.4 |
| 2023 Aso_Minamioguni 2 | 12 March 2023 | 114 | 80 | Quartz fiber | 9.1 |
| 2023 Aso_Ubuyama 1 | 20 March 2023 | 129 | 80 | Quartz fiber | 10 |
| 2023 Aso_Ubuyama 2 | 29 March 2023 | 75 | 65 | Quartz fiber | 4.9 |
| 2023 Aso_Minamioguni 3 | 5 March 2023 | 37 | 80 | Quartz fiber | 3.0 |
| 2023 Aso_Minamioguni 4 | 11 March 2023 | 123 | 80 | Quartz fiber | 9.8 |
| 2023 Aso_Ubuyama 3 | 29 March 2023 | 99 | 80 | Quartz fiber | 7.9 |
| 2023 Aso_Minamioguni 5 | 5 March 2023 | 132 | 80 | Quartz fiber | 11 |
| s2022 Aso_Oguni | 25 March 2022 | 219 | 83 | PTFE-coated glass fiber | 18 |
| 2022 Aso_Minamioguni | 13 March 2022 | 252 | 80 | PTFE-coated glass fiber | 20 |
| 2022 Aso city 1 | 6 March 2022 | 43 | 80 | PTFE-coated glass fiber | 3.4 |
| 2022 Aso city 2 | 6 March 2022 | 15.7 | 69 | PTFE-coated glass fiber | 16 |
| 2022 Background Kurokawa | 2 June 2022 | 150 | 80 | Quartz fiber | 12 |
| 2022 Background Waita | 3 June 2022 | 120 | 83 | Quartz fiber | 9.9 |
| 2023 Background Hirado | 4 February 2023 | 300 | 80 | Quartz fiber | 24 |
| 2023 Background Akiyoshidai | 12 February 2023 | 301 | 80 | Quartz fiber | 24 |

Meteorological information (average values) during the sampling is shown in Table 2. The information was obtained from the website of Japan Meteorological Agency

or JMA (https://www.data.jma.go.jp/obd/stats/etrn/select/prefecture00.php?prec_no=&block_no=&year=&month=&day=, accessed 25–30 January 2024) for the observatories at Hirado, Akiyoshidai, and Minamioguni (representing the Aso region). Wind directions did not coincide with typically observed directions (i.e., average values from the past 10 years' data, according to the JMA site) during the season (Table 3), but this is more likely due to the fact that Noyaki is conducted when the weather conditions are good, which is not typical weather in the region during that season. Similarly, the temperature in the Aso region during the studies (Table 2) was relatively higher than the typical temperature (Table 3).

**Table 2.** Meteorological information during the sampling.

|  | Temp (°C) | Wind Direction | Wind Speed (m s$^{-1}$) | Humidity (%) |
|---|---|---|---|---|
| 2023 Hirado | 8.2 | E | 1.8 | 50 |
| 2023 Akiyoshidai | 5.0 | W | 1.2 | 88 |
| 2023 Aso_Nishihara | 7.4 | NNE | 5.3 | na [‡] |
| 2023 Aso_Minamioguni 1 | 12.3 | NE | 2.1 | 42 |
| 2023 Aso_Minamioguni 2 | 18.1 | SSW | 2.1 | 58 |
| 2023 Aso_Ubuyama 1 | 17.3 | ESE | 2.8 | 47 |
| 2023 Aso_Ubuyama 2 | 14.1 | SE | 1.8 | 33 |
| 2023 Aso_Minamioguni 3 | 12.3 | NE | 2.1 | 42 |
| 2023 Aso_Minamioguni 4 | 18.1 | SSW | 2.1 | 58 |
| 2023 Aso_Ubuyama 3 | 14.1 | SE | 1.8 | 33 |
| 2023 Aso_Minamioguni 5 | 12.3 | NE | 2.1 | 42 |
| 2022 Aso_Oguni | 8.3 | N | 2.6 | na [‡] |
| 2022 Aso_Minamioguni | 17.6 | S | 2.4 | na [‡] |
| 2022 Aso city 1 | 8.4 | N | 5 |  |
| 2022 Aso city 2 | 8.2 | E | 1.775 | 50 |
| 2022 Background Kurokawa | 27.8 | SSW | 2.45 | na [‡] |
| 2022 Background Waita | 26.3 | NW | 2.1 | na [‡] |
| 2023 Background Hirado | 5.3 | NNE | 1.6 | 57 |
| 2023 Background Akiyoshidai | 13.3 | S | 2.1 | 59 |

[‡] Data are not available.

**Table 3.** Typically observed meteorological elements near the sampling sites [†].

|  | Temp (°C) | Monthly Precipitation (mm) | Prevailing Wind Direction | Wind Speed (m s$^{-1}$) | Humidity (%) |
|---|---|---|---|---|---|
| February Hirado | 7.6 | 93.6 | NW | 3.6 | 65 |
| February Akiyoshidai | 4 | 91.4 | na [‡] | 2.5 | na [‡] |
| March Minamioguni | 6.8 | 162 | S | 1.2 | na [‡] |

[†] The meteorological data were obtained from the website of Japan Meteorological Agency. [‡] Data are not available.

The extraction and analysis procedure for water-soluble ions via ion chromatography is well established and can be found elsewhere [12–17]. Briefly, each filter sample was placed in a wide-open-mouth 60 mL glass jar. Then, 10 mL of Milli-Q water (Integral 3, Merck KGaA, Darmstadt, Germany) was added to the jar and left until the sample filters were submerged in the water. The glass container was capped and sonicated for 15 min to extract inorganic components. A spike of ethanol to the filter sample is recommended

prior to extraction if PTFE filter media are used in order to reduce the hydrophobicity of the filter [12,14,15]. However, ethanol was not spiked to our PTFE-coated glass filter samples because the filters were easily submerged in Milli-Q water and, thus, PM collected on the filters sufficiently contacted with Milli-Q water. An aliquot of the extract was taken using a 1 mL disposable polypropylene syringe and then injected twice (a flushing and a sample injection) into the ion chromatograph (Dionex ICS-1000 or ICS-2000, Thermo Fisher Scientific, Waltham, MA, USA) through a Milli-Q-washed syringe filter (Whatman, Maidstone, UK, 25 mm o.d., 0.45 μm pore disc filter). The injection loops were 50 μL for cations and 25 μL for anions, respectively. For cation analysis of sodium, ammonium, potassium, magnesium, and calcium ($Na^+$, $NH_4^+$, $K^+$, $Mg^{2+}$, $Ca^{2+}$, respectively), a 4 mm suppressor (Dionex CDRS 600 4 mm, Thermo Fisher Scientific, Waltham, MA, USA), a guard column (Dionex IonPac$^{TM}$ CG12A, 12 × 50 mm, Thermo Fisher Scientific, Waltham, MA, USA), and an analytical column (Dionex IonPac$^{TM}$ CS12A, 4 × 250 mm, Thermo Fisher Scientific, Waltham, MA, USA) were used. For anion analysis of chloride, bromide, nitrate, phosphate, and sulfate ($Cl^-$, $Br^-$, $NO_3^-$, $PO_4^{3-}$, $SO_4^{2-}$, respectively), a 4 mm suppressor (Dionex ADRS 600 4 mm, Thermo Fisher Scientific, Waltham, MA, USA), a guard column (Dionex IonPac$^{TM}$ AG22 4 × 50 mm, Thermo Fisher Scientific, Waltham, MA, USA), and an analytical column (Dionex IonPac$^{TM}$ AS22 4 × 250 mm, Thermo Fisher Scientific, Waltham, MA, USA) were used. In both analyses, column temperature was set to 308 K, and the flow rate of the eluent was 1 mL min$^{-1}$. For cation and anion analyses, 20 mM methanesulfonic acid and a mixture of 4.5 mM $Na_2CO_3$/1.4 mM $NaHCO_3$ were used as eluents, respectively. The instrument was calibrated prior to the analysis using standard mixtures (6 cation mixed standard solution III and 6 anion mixed standard solution II, Kanto Chemical Co., Inc., Tokyo, Japan). Measurement was performed within a week of the sample extraction.

## 3. Results and Discussion

### 3.1. Blank Values, Detection Limits, and Background Concentrations

The typical measurement reproducibility of the analysis, estimated from the consecutive replicate measurements of sample extracts, was better than 6%. All sample extracts were pre-analyzed to determine their approximate concentrations. Following that, the extracts were diluted to optimal concentrations so that the obtained results were interpolated within the calibration ranges with the minimal memory effect from the previous run. Typically, concentrations of targeted ionic components in the chromatograms obtained by Milli-Q water injections right after the sample analysis, which gives an idea of the level of memory effect, are negligibly small. Based on the calibration curves created during the two analytical periods (four calibrations for the cation analysis, three calibrations for the anion analysis), detection limits (DLs) were determined for each chemical species, and the averaged values are summarized in Table 4. The average (±standard deviation, or SD) of seven BLK values for the analyzed chemical species ranged from 0.05 ± 0.02 to 2.6 ± 0.5 μg per filter for cations and from not detected (n.d.) to 0.17 ± 0.03 μg per filter for anions (Table 4). High blank-to-DL ratios for $Na^+$, $NH_4^+$, and $K^+$ implied some contamination issues (Table 4). The high ratios imply that there was some degree of contamination during the handling of samples (i.e., the procedures involving the TSP sampling and extract preparation). Our BLK values were then compared with the other BLK values reported [7]. Their blank values, given in μg m$^{-3}$, were converted to μg sample$^{-1}$, assuming that a volume of 71 m$^3$ was used for the determination of their atmospheric blank values. The comparison showed that their blank values were 1.4–284 times higher than our values. Their sampling filters and method were different from ours; thus, the difference in BLK values may be attributed to this difference in methodology. However, it is worth noting that our BLKs for $Na^+$ and $NH_4^+$ were comparable to the reported blank values; the ratios of the BLKs reported in the literature to our BLKs were only 1.4 and 2.3 times higher for these two chemical species, respectively, and for other species, the ratios were 32–284 times higher. Therefore, it should be stated that there may be a hidden significant impact of BLK concentrations on $Na^+$ and $NH_4^+$ concentrations.

It should be noted that we did not conduct recovery tests using reference materials because reference materials of airborne PM for such ionic chemical species were not available [12] until recently. However, our targeted chemical species are well known to be highly soluble in water, and extraction tests previously conducted with filtration and sonication methods did not show any significant difference [18]. The use of PTFE sampling filters may cause floating on the extracting solvent during the extraction, resulting in poor recovery [12]. However, this is not the case in our study, since sonicating extraction was performed, where the sample filters were completely submerged in the extracting solvent, Milli-Q water, in advance. In addition, sonication extraction has been applied to nearly all of the reported analyses for these ionic species in airborne PM to date, including PM from biomass burning [14,15], and there is no report on problems with sonication extraction. Overall, our analytical methodology is consistent with other studies; thus, the obtained results are comparable to the values in the literature.

Concentrations of atmospheric chemical species in the BKG samples ($n = 4$) varied widely (Table 4). The observed concentration ranges were 0.3–2.1, 0.16–0.85, 0.04–0.96, 0.04–0.08, and 0.14–0.38 $\mu g\ m^{-3}$ for $Na^+$, $NH_4^+$, $K^+$, $Mg^{2+}$, and $Ca^{2+}$, respectively. For $Cl^-$, $Br^-$, $NO_3^-$, $PO_4^{3-}$, and $SO_4^{2-}$, the ranges were 0.09–0.86, not detected (n.d.) for all samples, n.d.–4.0, n.d.–1.6, and 1.3–5.9 $\mu g\ m^{-3}$, respectively. Among the results, all $Br^-$ concentrations in the BKG samples were below the detection limit (b.d.l.), and three $PO_4^{3-}$ samples out of four were b.d.l. as well. The total masses of the chemical species in all BKG filter extracts, except one $NO_3^+$ concentration datum, were more than 3 times higher than their BLK values; thus, their quantitative analysis was valid. There was only one sample showing no detection of $NO_3^+$ quantity. Comparisons between our BKG concentration observations in the grasslands were compared to the reported concentrations of these chemical species in the Kyushu/Okinawa region.

**Table 4.** Averaged blank concentrations (BLK, μg per filter *, n = 7), instrumental detection limits (DL, μg per 10 mL solution or per filter), averaged atmospheric background concentrations (BKG, $\mu g\ m^{-3}$, n = 4) **, and reported atmospheric concentrations of ionic species.

| | $Na^+$ | $NH_4^+$ | $K^+$ | $Mg^{2+}$ | $Ca^{2+}$ |
|---|---|---|---|---|---|
| BLK (this study) | 2.6 ± 0.5 | 0.6 ± 0.2 | 0.2 ± 0.1 | 0.03 ± 0.02 | 1.2 ± 0.3 |
| DL (this study) | 0.04 ± 0.01 | 0.05 ± 0.04 | 0.02 ± 0.02 | 0.13 ± 0.05 | 0.3 ± 0.1 |
| BLK/DL (this study) | 65 | 12 | 10 | 0.2 | 5 |
| BKG (this study) | 1.2 ± 0.9 | 0.6 ± 0.3 | 0.4 ± 0.4 | 0.06 ± 0.02 | 0.2 ± 0.1 |
| Kumamoto/Nagasaki [#] | <0.05–0.6 | 0–9 | 0–2.5 | <0.01–0.17 | 0.01–0.13 |
| Fukuoka [°] | 0.023 | 0.018 | 0.039 | 0.024 | 0.040 |
| 9 locations in Khyushu/Okinawa | 1.0–5.7 | 0.9–3.6 | 0.18–0.48 | 0.14–0.99 | 0.48–3.1 |

| | $Cl^-$ | $Br^-$ | $NO_3^-$ | $PO_4^{3-}$ | $SO_4^{2-}$ |
|---|---|---|---|---|---|
| BLK (this study) | 0.05 ± 0.02 | n.d. [†] | 0.111 ± 0.004 | n.d. [†] | 0.17 ± 0.03 |
| DL (this study) | 0.05 ± 0.03 | 0.03 ± 0.02 | 0.11 ± 0.09 | 0.06 ± 0.20 | 0.13 ± 0.12 |
| BLK/DL (this study) | 1.0 | n.a. [§] | 1.0 | n.a. [§] | 1.3 |
| BKG (this study) | 0.48 ± 0.33 | 0 ± 0 [‡] | 1.6 ± 1.7 [‡] | 0.4 ± 0.8 [‡] | 2.8 ± 2.1 |
| Kumamoto/Nagasaki [#] | 0–1.4 | n.a. | 0–7 | n.a. | 0–24 |
| Fukuoka [°] | 0.036 | n.a. | 0.062 | n.a. | 0.096 |
| 9 locations in Khyushu/Okinawa [***] | 1.3–8.3 | n.a. | 0.05–9.5 | n.a. | 5.1–10.1 |

* Each filter sample was extracted with 10 mL Milli-Q water. ** All values are shown with ± SDs. [†] Not detected. [‡] n.d. was substituted with zero to determine averaged atmospheric concentrations. [§] Not applicable due to n.d. in the blank. [#] The concentration range data were for PM$_{2.5}$ in the summer through winter of 2013, and the values are estimated from the figures; "n.a." indicates data are not available [19]. [°] The concentration range data are for the averaged values of 1998–2012 PM observations in Dazaifu, Fukuoka [20]. [***] The concentration range data are for 24 h integrated TSP filter samples collected over an 11-day period in October 1991 at 9 locations in Fukuoka, Saga, Nagasaki, Oita, Kumamoto, Kagoshima, and Okinawa Prefectures [21].

Our observations of the BLK, DL, and BKG values are summarized in Table 4, together with some reported BKG values. Overall, the concentrations are still in the range of the concentrations reported by others, including the high blank concentration species of $Na^+$

and $NH_4^+$. There may be a critique that four samples may not be sufficient to define the average BKG values, and it is clear that observations over a long time period would have given us more representative BKG values. However, our BKG values are in the range of observable concentrations reported by others in this region, and no more BKG data are currently available. For these reasons, we accepted the averaged BKG concentrations here and applied the BKG values to evaluate the emission of water-extractable ionic species in TSP from the biomass burning.

### 3.2. Particulate Inorganic Substances Emitted from Grassland Burning

Blank-corrected concentrations of the ionic species showed that almost all values were positive and there was only one datum for $Ca^{2+}$ with a negative concentration (Table 5). That is, even the majority of total $Na^+$ and all $NH_4^+$ masses for each sample collected from Noyaki (i.e., the high blank chemical species) were significantly higher than the BLK values (4.5–16 and 11–250 times higher than the BLKs for $Na^+$ and $NH_4^+$, respectively). However, total $Na^+$ masses on the 5 PM filter samples were comparable to or slightly higher than the BLK value (i.e., 1.2–2.8 times higher than the BLK). Considering the background level of those ions, it should be noted that the data for those five samples need to be interpreted carefully.

Observed concentrations for each chemical species found in the water extracts from 15 TSP filter samples collected from Noyaki plumes showed large variations, except $Br^-$ and $PO_4^{3-}$ (Table 5). $Br^-$ was not detected in all samples, and $PO_4^{3-}$, which is a plant nutrient often found in soil and used as a fertilizer, was detected in only four samples, suggesting that open grassland burning was unlikely the source of those species in PM. However, some detections of $PO_4^{3-}$ imply a minor impact of soil dust blown upward during Noyaki. Meanwhile, all the other particulate inorganic ions in the air were elevated, indicating their emissions from Noyaki. Remarkably increased concentrations were observed for $NH_4^+$, $K^+$, $Ca^{2+}$, and $Cl^-$, followed by some extent of emissions for $Na^+$, $NO_3^-$, and $SO_4^{2-}$. To evaluate the emission of particulate inorganic ions, the BKG concentration for each species in Table 4 was subtracted from each of the observed concentrations shown in Table 5. This background-subtracted concentration is defined as an excess concentration and denoted using the symbol "$\Delta$". Many of the excess concentrations resulted in negative concentrations, suggesting their insignificant emissions from Noyaki (Table 6). We then evaluated their correlations.

**Table 5.** Observed atmospheric concentrations of chemical species found in TSP collected from Noyaki plumes *.

| Sample | $Na^+$ | $NH_4^+$ | $K^+$ | $Mg^{2+}$ | $Ca^{2+}$ | $Cl^-$ | $Br^-$ | $NO_3^-$ | $PO_4^{3-}$ | $SO_4^{2-}$ |
|---|---|---|---|---|---|---|---|---|---|---|
| 2023 Hirado | 0.6 | 1.2 | 1.1 | 0.98 | 1.8 | 0.14 | n.d. ‡ | 0.27 | n.d. ‡ | 0.12 |
| 2023 Akiyoshidai | 0.5 | 2.9 | 2.5 | 0.85 | 4.4 | 0.77 | n.d. ‡ | 0.62 | n.d. ‡ | 0.60 |
| 2023 Aso_Nishihara | 1.8 | 4.5 | 10.1 | 5.16 | 24.6 | 1.16 | n.d. ‡ | 0.28 | n.d. ‡ | 1.10 |
| 2023 Aso_Minamioguni 1 | 4.7 | 20.3 | 10.9 | 3.46 | 35.6 | 5.06 | n.d. ‡ | 0.67 | 0.3 | 1.75 |
| 2023 Aso_Minamioguni 2 | 1.5 | 10.6 | 6.5 | 2.32 | 23.4 | 0.15 | n.d. ‡ | 0.04 | 1.1 | 0.16 |
| 2023 Aso_Ubuyama 1 | 0.9 | 5.4 | 5.6 | 1.59 | 11.1 | 0.11 | n.d. ‡ | 0.03 | 0.1 | 0.09 |
| 2023 Aso_Ubuyama 2 | 0.4 | 4.2 | 4.6 | 1.45 | 9.0 | 0.80 | n.d. ‡ | 0.15 | n.d. ‡ | 0.51 |
| 2023 Aso_Minamioguni 3 | 0.2 | 2.0 | 0.6 | 0.46 | 0 † | 0.13 | n.d. ‡ | 0.42 | n.d. ‡ | 0.39 |
| 2023 Aso_Minamioguni 4 | 1.4 | 8.3 | 4.2 | 1.37 | 15.1 | 0.09 | n.d. ‡ | 0.02 | n.d. ‡ | 0.13 |
| 2023 Aso_Ubuyama 3 | 1.6 | 6.8 | 12.8 | 3.27 | 14.4 | 3.49 | n.d. ‡ | 0.82 | n.d. ‡ | 2.43 |
| 2023 Aso_Minamioguni 5 | 1.3 | 14.1 | 5.0 | 1.44 | 19.4 | 3.29 | n.d. ‡ | 0.48 | n.d. ‡ | 1.12 |
| 2022 Aso_Oguni | 1.6 | 0.6 | 0.8 | 0.25 | 2.0 | 0.55 | n.d. ‡ | 1.11 | n.d. ‡ | 2.44 |
| 2022 Aso_Minamioguni | 1.7 | 0.6 | 1.2 | 0.31 | 2.8 | 1.62 | n.d. ‡ | 0.52 | n.d. ‡ | 2.80 |
| 2022 Aso city 1 | 8.6 | 0.8 | 4.8 | 1.80 | 10.0 | 8.05 | n.d. ‡ | 1.41 | n.d. ‡ | 5.03 |
| 2022 Aso city 2 | 2.5 | 6.3 | 6.7 | 1.41 | 5.4 | 16.96 | n.d. ‡ | 1.74 | n.d. ‡ | 4.59 |

* According to the typical reproducibility of the measurements, a propagated uncertainty for each concentration is estimated to be 12%. † Concentration is treated as zero due to the blank-corrected concentration being negative in value. ‡ Not detected.

**Table 6.** Excess concentration (Δ) of chemical species found in TSP collected from Noyaki plumes.

| Sample | $\Delta Na^+$ | $\Delta NH_4^+$ | $\Delta K^+$ | $\Delta Mg^{2+}$ | $\Delta Ca^{2+}$ | $\Delta Cl^-$ | $\Delta NO_3^-$ | $\Delta SO_4^{2-}$ |
|---|---|---|---|---|---|---|---|---|
| 2023 Hirado | −0.6 | 0.6 | 0.8 | 0.92 | 1.5 | −0.34 | −1.32 | −2.69 |
| 2023 Akiyoshidai | −0.7 | 2.4 | 2.1 | 0.80 | 4.2 | 0.28 | −0.96 | −2.21 |
| 2023 Aso_Nishihara | 0.7 | 4.0 | 9.7 | 5.10 | 24.4 | 0.68 | −1.30 | −1.71 |
| 2023 Aso_Minamioguni 1 | 3.5 | 19.7 | 10.5 | 3.41 | 35.3 | 4.58 | −0.92 | −1.06 |
| 2023 Aso_Minamioguni 2 | 0.4 | 10.0 | 6.1 | 2.27 | 23.2 | −0.34 | −1.55 | −2.65 |
| 2023 Aso_Ubuyama 1 | −0.3 | 4.9 | 5.2 | 1.53 | 10.8 | −0.37 | −1.56 | −2.71 |
| 2023 Aso_Ubuyama 2 | −0.7 | 3.7 | 4.2 | 1.39 | 8.7 | 0.32 | −1.44 | −2.30 |
| 2023 Aso_Minamioguni 3 | −0.9 | 1.5 | 0.2 | 0.40 | −0.6 | −0.36 | −1.16 | −2.42 |
| 2023 Aso_Minamioguni 4 | 0.3 | 7.7 | 3.8 | 1.31 | 14.9 | −0.40 | −1.56 | −2.68 |
| 2023 Aso_Ubuyama 3 | 0.4 | 6.3 | 12.4 | 3.21 | 14.2 | 3.00 | −0.76 | −0.38 |
| 2023 Aso_Minamioguni 5 | 0.1 | 13.6 | 4.6 | 1.38 | 19.2 | 2.81 | −1.10 | −1.69 |
| 2022 Aso_Oguni | 0.5 | 0.0 | 0.4 | 0.19 | 1.8 | 0.06 | −0.48 | −0.37 |
| 2022 Aso_Minamioguni | 0.5 | 0.0 | 0.8 | 0.25 | 2.6 | 1.13 | −1.06 | 0.00 |
| 2022 Aso city 1 | 7.5 | 0.2 | 4.4 | 1.74 | 9.7 | 7.57 | −0.18 | 2.23 |
| 2022 Aso city 2 | 1.3 | 5.7 | 6.3 | 1.35 | 5.2 | 16.48 | 0.15 | 1.78 |

Correlations between the chemical species showed some highly correlated species, such as $K^+$, $Mg^{2+}$, and $Ca^{2+}$ (Table 7). Potassium has been suggested as an indicator of biomass burning PM [6]; therefore, the high correlations here (e.g., Figure 2) may suggest the emission of particulate magnesium and calcium during Noyaki. Meanwhile, anion species and $Na^+$ were not as highly correlated as cations were with potassium ($r^2 < 0.09$). However, high correlations were still observed between the anion species ($r^2 = 0.64$–$0.83$), regardless of the negative excess concentrations (Table 6) and of the negative intercepts at x- or *y*-axis in their correlations (e.g., Figure 3). We suspect that those species were possibly emitted from Noyaki, but their relatively high BLK and BKG concentrations may have concealed the actual emissions. With the current dataset, we still could not exclude a possibility that the emissions of these chemical species are low due to their low contents in the fuel (i.e., the major plant species of *Miscanthus sinensis* and *Pleioblastus chino var. viridis*) and/or locations. More careful sample handling and analysis for minimal blank concentrations, collecting a greater number of BKG and Noyaki samples in a variety of locations, as well as conducting analysis of chemical contents in plant species will allow us to evaluate the significance of grassland burning as a source of highly correlated ions. Thus, it will be our future work.

**Table 7.** Coefficient of determination between ionic species.

| | $\Delta Na^+$ | $\Delta NH_4^+$ | $\Delta K^+$ | $\Delta Mg^{2+}$ | $\Delta Ca^{2+}$ | $\Delta Cl^-$ | $\Delta NO_3^-$ | $\Delta SO_4^{2-}$ |
|---|---|---|---|---|---|---|---|---|
| $\Delta Na^+$ | 1 | 0.014 | 0.074 | 0.063 | 0.087 | 0.261 | 0.317 | 0.517 |
| $\Delta NH_4^+$ | | 1 | 0.357 | 0.208 | 0.718 | 0.020 | 0.024 | 0.027 |
| $\Delta K^+$ | | | 1 | 0.783 | 0.589 | 0.089 | 0.005 | 0.024 |
| $\Delta Mg^{2+}$ | | | | 1 | 0.628 | 0.008 | 0.011 | 0.000 |
| $\Delta Ca^{2+}$ | | | | | 1 | 0.000 | 0.055 | 0.016 |
| $\Delta Cl^-$ | | | | | | 1 | 0.683 | 0.641 |
| $\Delta NO_3^-$ | | | | | | | 1 | 0.836 |
| $\Delta SO_4^{2-}$ | | | | | | | | 1 |

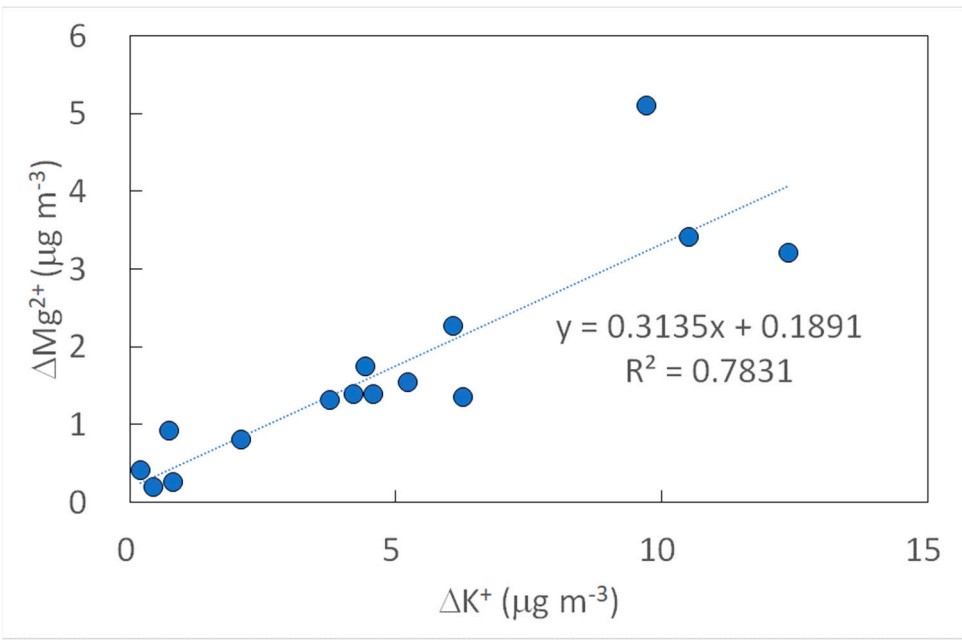

**Figure 2.** Scatter plot of excess atmospheric concentration (Δ) of potassium and magnesium ions in TSP collected from Noyaki plumes.

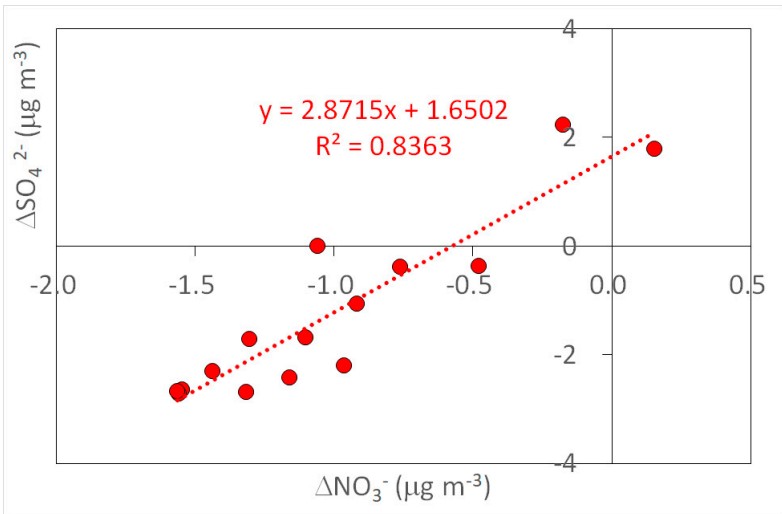

**Figure 3.** Scatter plot of excess atmospheric concentration (Δ) of nitrate and sulfate ions in TSP collected from Noyaki plumes.

*3.3. Comparison with Other Biomass Burning Emission Studies*

Our observations were compared with values reported in the literature, as shown in Table 8. It has been widely recognized that most biomass burning emits potassium [22]. Our observations also agreed with this finding. Ammonium ions have been also recognized as particulate substances emitted from biomass burning, which agrees with our observations as well. A unique feature we found was that there were substantial amounts of calcium and magnesium, but insignificant emissions of chloride ions. Since phosphate was not detected in our samples, the influence of soil dust in the grasslands was not relatively important (relative to the PM from grassland burning). Our contradicting observations of the high correlation between nitrate and sulfate and their insignificant excess emissions can be explained by the fact that those substances are actually emitted from grassland burning, which agrees with the values reported in the literature shown in Table 8, and the high BLK

and BKG made the location of the scatter plot in Figure 3 shift. More careful analysis of these ubiquitous elements may lead us to more convincing evidence of their emissions.

**Table 8.** Comparison of inorganic ion concentrations found in biomass-burning plumes and smoke and concentration profiles used in a chemical mass balance (CMB) model.

| Sample | Na$^+$ | NH$_4^+$ | K$^+$ | Mg$^{2+}$ | Ca$^{2+}$ | Cl$^-$ | NO$_3^-$ | SO$_4^{2-}$ |
|---|---|---|---|---|---|---|---|---|
| This study [**] | 1.0 | 5.3 | 4.8 | 1.7 | 12 | 2.6 | 0.22 | 0.46 |
| California chaparral (mass %) [***] | 0.33 | 0.82 | 4.86 | n.a. * | n.a. * | 4.15 | 1.34 | 2.2 |
| Oregon rye grass (mass %) [***] | 0.24 | 0.16 | 0.89 | n.a. * | n.a. * | 1.17 | 0.11 | 1 |
| Burning period (μg m$^{-3}$) [†] | 0.8 | 0.8 | 1.2 | n.a. * | 1.9 | 0.8 | n.a. * | 3.2 |
| Smoke samples (average mass fraction) [‡] | 0.0235 | n.a. * | 0.0775 | n.a. * | n.a. * | 0.1075 | n.a. * | 0.025375 |
| BB plumes in Asia (μg m$^{-3}$) [∥] | n.a. * | 0.398 | n.a. * | n.a. * | n.a. * | 0.0192 | 0.138 | 1.67 |
| Biomass burning (μg m$^{-3}$) [§] | n.a. * | 1 | 0.11 | 0.005 | 0.01 | <0.01 | 0.15 | 2 |
| 10.6% moist. rice straw (g kg$^{-1}$) [⌐] | n.a. * | 0.083 | n.a. * | n.a. * | n.a. * | 0.3 | 0.006 | 0.027 |
| 12% moist. wheat straw (g kg$^{-1}$) [⌐] | n.a. * | 0.034 | n.a. * | n.a. * | n.a. * | 0.12 | 0.006 | 0.031 |
| 11.2% moist. barley straw (g kg$^{-1}$) [⌐] | n.a. * | 0.081 | n.a. * | n.a. * | n.a. * | 1.53 | 0.005 | 0.121 |
| 13.1% moist. rice husk (g kg$^{-1}$) [⌐] | n.a. * | 0.043 | n.a. * | n.a. * | n.a. * | 0.13 | 0.007 | 0.02 |
| Dry (biomass burning) period (μg m$^{-3}$) [⌐] | n.a. * | 1.41 | 0.95 | n.a. * | n.a. * | n.a. * | 2.23 | 1.44 |
| Average agricultural crop (mass %) | 0.35 | 0.79 | 8.9 | 0.063 | 0.55 | 4.9 | 0.37 | 2.5 |
| PM$_{2.5}$ dry season in Tanzania (μg m$^{-3}$) | 0.62 | 0.93 | 1.5 | 0.079 | 0.30 | 0.08 | 0.18 | 0.26 |
| PM$_{10}$ dry season in Tanzania (μg m$^{-3}$) | 2.2 | 0.65 | 1.9 | 0.46 | 1.7 | 0.33 | 0.44 | 0.56 |
| Fine PM from fresh smoke during SAFARI 2000 (μg m$^{-3}$) | n.a. * | n.a. * | 4.5 | n.a. | 0.10 | 0.27 | 1.4 | 1.9 |
| Coarse PM from fresh smoke during SAFARI 2000 (μg m$^{-3}$) | n.a. * | n.a. * | 1.4 | 0.64 | 2.2 | 0.26 | 4.2 | 1.3 |
| CMB biomass burning profile (unitless) [#] | 33 | n.a. * | 89 | n.a. * | 4 | 100 | n.a. * | n.a. * |

* n.a.: Not available. [**] Reported values are average values, and negative values were treated as zero for averaging. [***] Values from Rogers et al. [23]. [†] Values from Ezcurra et al. [24]. [‡] Values from Novakov et al. [25]. [∥] Values from Kondo et al. [26]. [§] Values from Aggarwal et al. [27]. [⌐] Values from Hayashi et al. [28]. [⌐] Values from Fuzzi et al. [29]. [#] Values from Maenhaut et al. [30]. Values from Park et al. [31]. Values from Mkoma et al. [16]. Values from Formenti et al. [32].

## 4. Conclusions

Emissions of particulate inorganic substances were evaluated in this study. The observations showed an elevation in the concentrations of NH$_4^+$, K$^+$, Mg$^{2+}$, and Ca$^{2+}$, while the elevation of the concentrations of anion species and Na$^+$ was not as high as that of cations. Taking potassium as an indicator of biomass burning, magnesium and calcium were highly correlated, suggesting that these three cation species were emitted from the Noyaki in this region as well. Meanwhile, correlations between anions, as well as between cation (except Na$^+$) and anion species, were not as high as correlations between cations. These poor correlations may be attributed to high BKG concentrations; therefore, further investigation will be needed, combined with more careful sample handling and chemical analysis. Long-term observations on BKG concentrations will characterize the BKG in this region well, consequently allowing us to more precisely characterize the emissions of these chemical species from Noyaki. The information characterized in studies on regional and annual prescribed burning events can potentially be extended to evaluations of the levels of fine PM (PM$_{2.5}$) with regard to their short- and mid-term adverse health effects. In this case, it will be worthwhile to expand the target species, including PM mass concentrations and size distribution, as well as toxic chemical species, such as polycyclic aromatic hydrocarbons [33].

**Author Contributions:** Conceptualization, S.I.; Sample collection, S.I., S.K., A.S., H.S. and S.Y.; Sample analysis, S.I.; Preparation of manuscript, S.I., S.Y. and S.K. All authors have read and agreed to the published version of the manuscript.

**Funding:** This research was externally funded by The Sumitomo Foundation (Grant number 2230266) and internally funded by the National Institute for Minamata Disease or NIMD (Grant ID RS-23-11).

**Institutional Review Board Statement:** Conducting this research and the submission of this manuscript were approved by the director board of NIMD.

**Informed Consent Statement:** Not applicable.

**Data Availability Statement:** The data are not publicly available currently, however, we will provide those when requested.

**Acknowledgments:** The authors also thank the assistant Nanami Yamamoto from the NIMD for assisting our laboratory work in this project.

**Conflicts of Interest:** The authors declare no conflict of interest.

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
