# Peer review of "Emission of Particulate Inorganic Substances from Prescribed Open Grassland Burning in Hirado, Akiyoshidai, and Aso, Japan"

_2813-4168, doi:10.3390/air2010004_

Round 1

Reviewer 1 Report

Comments and Suggestions for Authors

Comments on the Quality of English Language

really need to be improved

Author Response

See the attached file for our responses.

Reviewer 2 Report

Comments and Suggestions for Authors

The manuscript begins by highlighting the importance of biomass burning as a major source of particulate matter emissions globally. However, the specific emission of particulate inorganic species from prescribed grassland burning in Japan has not been characterized yet. The authors then describe their study where they collected total suspended particulate matter from filters during prescribed grassland burning in Hirado, Akiyoshidai, and Aso, Japan. The collected samples were analyzed for water-soluble inorganic components using ion chromatography.

The measurement results showed high concentrations of potassium, calcium, and magnesium, which align with findings in the literature. However, the concentrations of sodium, chloride, nitrate, and sulfate were found to be insignificant, contrary to other biomass burning studies. A positive correlation between sulfate and nitrate concentrations was still observed.

It is suggested that the low concentrations of these species may have been biased in the measurements, possibly due to subtracting blank and background values. The authors propose that further data collection in this study may enable a better understanding of the significance of domestic biomass burning in contributing to particulate inorganic components in the Japanese air.

Overall, the manuscript provides valuable insights into particulate inorganic species emissions from prescribed grassland burning in Japan. However, the concerns raised regarding the measurements and the need for more data should be addressed in order to strengthen the conclusions.

1. Clarify the purpose and objectives of the study in the manuscript's introduction. It should clearly state the research gap that the study aims to fill by characterizing the emission of particulate inorganic species from prescribed grassland burning in Japan.

2. Provide more details about the methodology used for collecting the total suspended particulate matter from filters during prescribed grassland burning. Explain the sampling locations, duration, and any specific considerations taken during the collection process.

3. Elaborate on the ion chromatography analysis method used to determine the water-soluble inorganic components in the collected samples. Include information on the calibration procedures and any quality control measures taken during the analysis.

4. Discuss the potential reasons for the observed low concentrations of sodium, chloride, nitrate, and sulfate in the prescribed grassland burning emissions compared to other biomass burning studies. Consider factors like regional differences, variability in burning conditions, or specific vegetation types burned in the studied areas.

5. Address the concern raised regarding the bias in the measurements due to subtracting blank and background values. Explain how these values were determined and provide justification for their subtraction from the observed concentrations.

6. Emphasize the need for further data collection in the study to better understand the significance of domestic biomass burning in contributing to particulate inorganic components in the Japanese air. Discuss potential strategies for obtaining additional data and suggest how these additions could strengthen the conclusions.

7. Consider including a discussion on the potential implications of the findings, both in terms of air quality and public health. Relate the specific findings to the broader context of particulate matter pollution and its impact on human health and the environment.

8. Revise the conclusion to summarize the key findings and their implications. Address the concerns raised in the review regarding the measurements and the need for more data, and emphasize the potential future research directions in this field.

Overall, the manuscript's content is valuable, but these improvements will help strengthen the clarity and impact of the study.

9.    There are lots of commas and full stops are missing.

10.  References: There are fewer numbers of new references or recent references especially related to Emission of particulate inorganic substances studies. Therefore, for current study should be given a strong impact if you can cite the following reference.

a.    The Health Risk and Source Assessment of Polycyclic Aromatic Hydrocarbons (PAHs) in the Soil of Industrial Cities in India

b.    Long-term temperature trend analysis associated with agriculture crops

c.     Characterization and health risk assessment of particulate bound polycyclic aromatic hydrocarbons (PAHs) in indoor and outdoor atmosphere of Central East India

d.    Temporal variability of atmospheric particulate-bound polycyclic aromatic hydrocarbons (PAHs) over central east India: sources and carcinogenic risk assessment

e.    Deposition of trace metals associated with atmospheric particulate matter: Environmental fate and health risk assessment

f.      Emission sources, Characteristics and risk assessment of particulate bound Polycyclic Aromatic Hydrocarbons (PAHs) from traffic sites

Author Response

See the attached file for our replies.

Reviewer 3 Report

Comments and Suggestions for Authors

This is an interesting study on Emission of particulate inorganic substances from open grassland burning prescribed in Hirado, Akiyoshidai, and Aso, Japan. It is a good study, as revised, and merits publication in Air once a number of issues are addressed. More specifically:

1. Line 43: The word “limitation” does not seem appropriate here. Maybe “understanding”?

2. In section 2.1. more information should be provided on the season and dates of Noyaki, and on the prevailing meteorological conditions in Japan during that period, such as temperatures, prevailing wind direction, wind speeds, humidity, etc..

3. Lines 160-162 “Therefore, it should be stated that there may be hidden significant impact of the BLK concentrations on the Na+ and NH4+ concentrations.”: Further explanations should be provided here. Also, this statement seems a bit contradictory to the paragraph that followed it. Further links should be provided between these two paragraphs, as appropriate.

Comments on the Quality of English Language

Minor editorials needed.

Author Response

This is an interesting study on Emission of particulate inorganic substances from open grassland burning prescribed in Hirado, Akiyoshidai, and Aso, Japan. It is a good study, as revised, and merits publication in Air once a number of issues are addressed.

Authors’ reply: Thank you again for your interest and reviewing the manuscript.

More specifically:

  1. Line 43: The word “limitation” does not seem appropriate here. Maybe “understanding”?

Authors’ reply: We replaced the word “limitation” to goal.

  1. In section 2.1. more information should be provided on the season and dates of Noyaki, and on the prevailing meteorological conditions in Japan during that period, such as temperatures, wind direction, wind speeds, humidity, etc..

Authors’ reply: We added the sampling information in the new table (Table 1 in the revised manuscript). Although we did not measure any meteorological elements, we obtained the meteorological information for the nearest weather observatories that Japan Meteorological Agency possessed. The information is open in the internet. The meteorological information during the samplings was summarized in Table 2 and typical meteorological information in February and March is now given in Table 3 (i.e., new tables were added).

  1. Lines 160-162 “Therefore, it should be stated that there may be hidden significant impact of the BLK concentrations on the Na+ and NH4+ concentrations.”: Further explanations should be provided here. Also, this statement seems a bit contradictory to the paragraph that followed it. Further links should be provided between these two paragraphs, as appropriate.

Authors’ reply: Thanks for your comment. The evaluation of emission from Noyaki needs to consider not only the blank values, but background values. Even if concentrations were remained positive after deduction of the blank values, the values may likely be in negative when deducting the background values. We will need more extensive studies to characterize the blank and background values. This will lead us to a clearer clue to find where the problem exist or the emission is really insignificant. More studies are needed.

Round 2

Reviewer 1 Report

Comments and Suggestions for Authors

Comments to the Author

It was my pleasure to review the manuscript entitled “Title Emission of particulate inorganic substances from open grassland burning prescribed in Hirado, Akiyoshidai, and Aso, Japan” (air-2729930-v3)

Irei, S., et al.,

Comments:

1.       The typesetting is still a question. Like Table 2, there has blank line, and it should be cited first and then listed in the text. Table 3 was not cited in the text.

Line 221-223 were blank.

The data in tables did not have the consistent significant figure.

Line 266, figure 4 should be figure 3.

Table 5, 6 then should be followed by 7 and 8, not 4 and 5.

Table 5 and 6 should be consistent in format.

2.       In “materials and Method” only 2.1 sample collection, where is 2.2…...

Even though the analysis procedure was described here, the QA/QC did not detail.

3.       The whole text is still a bit mess.

Comments on the Quality of English Language

It is really should be improved. 

Round 3

Reviewer 1 Report

Comments and Suggestions for Authors

It can be accepted. 

Comments on the Quality of English Language

It can be improved.